# Corrosion Fatigue Fracture Characteristics of FSW 7075 Aluminum Alloy Joints

**DOI:** 10.3390/ma13184196

**Published:** 2020-09-21

**Authors:** Qingna Ma, Fei Shao, Linyue Bai, Qian Xu, Xingkun Xie, Mei Shen

**Affiliations:** College of Field Engineering, Army Engineering University of PLA, Nanjing 210007, China; mqingna1980@sina.com (Q.M.); q1058427910@sina.com (Q.X.); jfxie2020@sina.com (X.X.); shenmei2020@sina.com (M.S.)

**Keywords:** 7075 aluminum alloy, FSW joint, corrosion fatigue, microscopic structure, fracture characters

## Abstract

The corrosion fatigue properties and fracture characteristics of friction stir welding joints of 7075 aluminum alloys were studied via corrosion fatigue tests, electrochemical measurements, and corrosion fatigue morphology and microstructure observations. The results show that the corrosion fatigue crack of the friction stir welding (FSW) joint of 7075 aluminum alloys originated in the junction zone between the thermomechanically affected zone and the weld nugget zone. The corrosion fatigue life of the joint decreased with increasing stress amplitude, with an S–N curve equation of lgN = 5.845 − 0.014S. Multiple crack sources were observed in the corrosion fatigue fracture. The main crack source originated from the corrosion pits at the interface between the thermomechanically affected zone and the weld nugget zone due to the influence of the coarse microstructure and the large potential difference between both zones. Corrosion morphologies of a rock candy block and an ant nest appeared in the crack propagation zone and the grain boundary of the weld nugget zone. In addition, fatigue speckles and intergranular fractures were observed, as well as brittle fracture characterized by cleavage steps and secondary cracks in the final fracture zone.

## 1. Introduction

The 7075 aluminum alloys are widely used in marine engineering and other fields due to their good plasticity, high specific strength, and high specific stiffness. However, aluminum alloys are prone to local corrosion in marine environments due to the permeation of chloride ions, which reduces their fatigue life and leads to early failure [1,2]. The connection mode of aluminum alloy structures is mostly by welding. Because aluminum alloys have a low melting point, a high coefficient of thermal conductivity, and a high specific heat capacity, traditional fusion welding easily generates defects, such as hot cracking, porosity, and deformation, among others [3,4]. Friction stir welding (FSW) is a solid-state joining technology invented at TWI (The Welding Institute) in 1991 in the United Kingdom [5]. Such a process has the advantages of low residual stress and welding temperature, and it does not exhibit distortion of the welding parts [6,7]. This method, therefore, results in higher-quality structures of aluminum alloys than melting welding and has become the main welding method for high-strength aluminum alloys.

In marine environments, aluminum alloy structures are prone to pitting corrosion, intergranular corrosion, and exfoliation corrosion behaviors due to the adsorption and osmosis of corrosion factors (for example, Cl^−^) [8,9,10,11]. More specifically, the corrosion of welded joints [10,11,12,13] seriously affects the safe and reliable use of aluminum alloy structures. Therefore, in order to prevent the corrosion failure of welded aluminum structures in marine environments, great efforts have been directed toward the investigation of welded-joint corrosion. For example, the corrosion resistance of friction stir welding (FSW) joints was examined in different areas [12], as well as their corrosion behavior and corrosion mechanism [14,15,16,17,18]. However, aluminum alloy structures of ships and other marine engineering structures in corrosive environments are not only affected by corrosion, but also by the alternating load of wind and waves. Therefore, aluminum alloy structures are prone to corrosion fatigue failure due to the corrosive environment and cyclic loading. Consequently, it is necessary to understand the corrosion fatigue of aluminum alloy structures in the weakest position, i.e., the welded joint.

On one hand, the corrosion fatigue of aluminum alloy FSW joints can cause mechanical fatigue damage under cyclic stress. On the other hand, the electrochemical reaction between the corrosion medium and the aluminum alloy results in corrosion damage. When these two kinds of damages coexist, the corrosion effect of the corrosive medium on the crack-tip metal significantly increases the corrosion fatigue crack growth rate of the welded joint surface, leading to the premature failure of the FSW joint. However, current research on the corrosion fatigue of FSW joints mainly focuses on the corrosion fatigue study of samples after pre-corrosion combined with fatigue behavior [19,20,21,22,23], which is different from the coupled effect of the corrosion environment and the fatigue of structures under alternating loads in the oceanic environment. At present, there are few reports on the corrosion fatigue of FSW joints under the coupled effect of corrosion and fatigue, which is not enough to provide significant guidance for the improvement of the corrosion fatigue performance of welded joints. Therefore, it is necessary to study the corrosion fatigue performance of FSW joints of aluminum alloy under marine environments.

In this paper, the corrosion fatigue, S–N curve, and corrosion fatigue performance of 7075-T6 aluminum alloy FSW butt-welding joints were studied. The observation of the surface microstructure of the FSW joints and the corrosion fatigue fracture, the distribution of the joint region, and the joint region of the corrosion fatigue fracture were determined. The sensitive region of the joint corrosion was determined by the polarization curve of each zone obtained from electrochemical corrosion tests. According to the observation of the fracture morphology of the crack source region, the crack propagation region, and the transient fracture region, the fracture characteristics were revealed.

## 2. Materials and Experiments

### 2.1. Materials and Welding Process

A 10 mm thick 7075-T6 aluminum alloy plate made by Hebei Weihan Aluminum Company in Cangzhou, China was used in the present work. The chemical composition is shown in Table 1*,* and its mechanical properties are shown in Table 2. In Table 2, R_m_ is the tensile strength of 7075-T6 aluminum alloy. R_p0.2_ denotes the conditional yield strength of 7075-T6 aluminum alloy, which refers to the stress value of 0.2% nonproportional elongation. The joints were manufactured using predetermined welding parameters, as summarized in Table 3. The aluminum alloy plates with a thickness of 10 mm consisted of FSW butt joints. The dimensions of the aluminum alloy plates were 175 mm × 150 mm. The FSW plate is shown in Figure 1.

### 2.2. Corrosion Fatigue Test

The samples used in the corrosion fatigue test were linearly cut along the vertical direction of the weld center of the butt-welding plate. Because the upper surface of weld was uneven after friction stir welding, a thickness of 3 mm was polished from the upper surface using silicon carbide paper up to 1200 grit. In addition, to avoid the influence of the incomplete welding of the lower surface on the specimen, a thickness of 5 mm was polished from the lower surface. The position of the sample is shown in Figure 2, marked with a red line. Then, the dog-bone samples were linearly cut. The dog-bone sample used for the test is shown in Figure 3. The samples were divided into five groups before the test, corresponding to five different stress amplitudes, with three specimens in each group. The groups were labeled A to E, and each specimen in a group was numbered 1–3, obtaining groups such as A1, A2, A3, B1, …, E3. The machine used for performing the fatigue corrosion test was the corrosion fatigue testing system (CFS) (Figure 4) which made by Cor-Force company in Shanghai, China. Corrosion fatigue tests were performed in a stress-controlled mode with a stress ratio of 0.06 and a frequency of 3 Hz servo-hydraulic at ambient temperature. The load waveform was a sine wave. During the test, the reaction still was lowered to the bottom plate (Figure 4), and 3.5 wt.% NaCl solution was injected into the reaction still to immerse the sample completely.

### 2.3. Analysis of the Microstructure

Optical microscopy made by Olympus in Bavaria, Germany, was used for microstructure observation. Two different samples were analyzed, FSW metallographic specimens and corrosion fatigue fracture specimens. The dimensions of the FSW metallographic samples were 10 mm × 10 mm × 3 mm, which were cut by wire-cutting perpendicular to the weld center. Each specimen was etched by Keller’s reagent (1 mL of HF + 1.5 mL of HCl + 2.5 mL of HNO_3_ + 95 mL of H_2_O) for microstructural examination, after having its surface burnished. If polishing the corrosion fatigue fracture directly, both the fracture morphology features and the position of the corrosion fatigue fracture would change. Thus, in order to maintain the accurate position of the fracture, the upper and lower surfaces of the fracture were treated with the same method to observe the microstructure. Bilateral fractures were taken for observation to facilitate polishing, observing the upper surface of one side and the lower surface of the other side. The position of the corrosion fatigue fracture was determined by comparing the observation results of the microstructure before and after corrosion fatigue.

### 2.4. Electrochemical Corrosion Analysis

The electrochemical corrosion test consisted of an open-circuit potential (OCP) and potentiodynamic polarization curve. Electrochemical measurements were carried out on a three-electrode electrolyte cell system with a platinum electrode as the auxiliary electrode and a saturated calomel electrode as the reference electrode. The working electrodes were the weld nugget zone (WNZ), the thermomechanically affected zone (TMAZ), and the heat-affected zone (HAZ). All measurements were carried out at ambient temperature. The experimental instrument of electrochemical analysis was by AMETEK, Oak Ridge, TN, USA. The dimensions of the electrochemical corrosion samples were 50 mm × 50 mm × 10 mm. The area of the exposed surface was 1 cm^2^, and the remaining surface was sealed with epoxy. The working electrodes were polished with sandpaper of 400#, 600#, 800#, 1000#, 1200#, 1500#, and 2000# in series, and then polished with a polishing machine until the surfaces were as bright as a mirror. To ensure the working electrode reached a stable potential in solution, the OCP tests were conducted after being immersed in 3.5 wt.% NaCl solution for 30 min. Then, the scanning range of the polarization curve was determined. The potentiodynamic polarization curve was measured at a scanning rate of 3 mV/s.

### 2.5. Morphological Characteristics of the Fracture

The characteristics of the corrosion fatigue fracture were observed using a KYKY-EM6900 scanning electron microscope made by Beijing Zhongke Scientific Instrument Company of Beijing, China. The samples consisted of corrosion fatigue fracture specimens. The length of the specimens was lower than 10 mm, which were linearly cut from the corrosion fatigue fracture specimens and subsequently immersed for 10 min at a constant temperature of 90 °C in a water bath. After immersion, the fracture corrosion products were cleaned with a solution of 5 mL of H_3_PO_4_, 2 g of CrO_3_, and 100 mL of H_2_O, and ultrasonically cleaned with alcohol. After removing the corrosion products, the samples were put into test tubes filled with anhydrous ethanol for analyzing the characteristics of the corrosion fatigue fracture.

## 3. Results and Discussion

### 3.1. Corrosion Fatigue Tests

The corrosion fatigue tests were carried out in 3.5 wt.% NaCl solution, and the corrosion fatigue lives were obtained at different load levels, as shown in Table 4. All samples were fractured in the FSW joint regions, and the exact locations of the fracture were determined by subsequent microscopic structure observation.

According to the results of the corrosion fatigue test shown in Table 4, the S–N curve of the FSW joint corrosion fatigue was fitted using the least squares method, as shown in Figure 5. The results indicate that the corrosion fatigue life decreased greatly with increasing stress amplitude. The fitted S–N curve equation is shown in Table 5, where S is the load stress range, N represents the corrosion fatigue life, and m and C are material constants.

### 3.2. Microstructural Characterization

#### 3.2.1. Microstructure of the FSW Joint

The microstructure of the aluminum alloy changed during the FSW process. Three distinct zones, namely, WNZ, TMAZ, and HAZ, were generated due to the nonuniform heating and plastic flow [22,23,24]. The macrostructure of the FSW joint cross-section is shown in Figure 6, and the microstructure of the FSW joint is shown in Figure 7. The three main regions can be clearly distinguished, from inside to outside, as WNZ, TMAZ, and HAZ. WNZ was the most seriously affected by the thermal cycle and the mechanical action, and severe plastic deformation occurred under the influence of the high-temperature heat input and the strong shearing action of the stirring needle. The dynamic recrystallization phenomenon of WNZ resulted in the formation of small equiaxial microstructure [25]. The average grain size was approximately 5 µm. Additional second-phase particles were observed in grain boundaries and crystals [26] (Figure 8a). TMAZ produced elongated or curved tissue at a high-speed rotation of the friction stir head. In the FSW process, despite the dual effect of thermal cycling and mechanical action, TMAZ was far away from the stirring needle, and the stirring force suffered was far less than that of WNZ, where the deformation stress was not enough to result in recrystallization. Therefore, the grain size of this zone was large, and bending deformation occurred (Figure 8b). HAZ mainly underwent thermal cycling, without strong action or plastic deformation, forming a thick strip grain structure at low temperature [27] (Figure 8c).

#### 3.2.2. Microstructure of the Upper and Lower Surfaces of the Corrosion Fatigue Fracture

In order to determine the location of the corrosion fatigue fracture specimens, the microstructure of both sides of the corrosion fatigue fracture specimen was observed. Samples smaller than 10 mm were cut from both sides after corrosion fatigue fracture (as shown in Figure 9 in red boxes). The upper surface was selected on one side and the lower surface on the other side of the fracture (as shown on the disc by the white rectangle of Figure 9, left). Three zones of the upper and lower surface were analyzed as marked a, b, and c in Figure 9. Figure 10a shows that most of the grains on the upper surface of area a consisted of equiaxial crystals, and the distribution of grains showed a curved shape, with the microstructure characteristics of WNZ and TMAZ. This indicates that area a of the upper surface constituted the boundary zone between TMAZ and WNZ. Area b of the upper surface consisted of grains with large bending deformation, as shown in Figure 10b, with a TMAZ structure. Last, area c of the upper surface exhibited a small portion of fine bending deformation grains and most fine equiaxial grains (Figure 10c), which arose from the junction zone between TMAZ and WNZ to WNZ. Areas a, b, and c of the lower surface all consisted of WNZ, isoaxial crystals (Figure 10d–f). In the process of corrosion fatigue, the samples were subjected to the coupled action of corrosion and fatigue. The corrosion fatigue was located at the junction area of the upper (or lower) surface and the side of the sample, affected by stress concentration and corrosion. That is, the source could be located at the four angles of the fracture. However, since the corrosion resistance of the four corners could not be determined, the location of the corrosion fatigue source could not be accurately determined until the analysis of the electrochemical corrosion test results. According to the microstructure, the corrosion fatigue crack finally broke in the WNZ.

### 3.3. Local Electrochemical Measurements

Aluminum alloy is sensitive to corrosion in a solution containing chloride ions. In addition, corrosion substantially affects the initiation of the fatigue crack. The crack source due to general corrosion fatigue is mainly generated in the corrosion pits of the sample surface. In order to determine the location of the corrosion fatigue source, electrochemical corrosion tests were performed.

Figure 11 shows the OCP for the WNZ, TMAZ, and HAZ in 3.5 wt.% NaCl solution. The open-circuit potential of the WNZ, TMAZ, and HAZ was −0.803 V, −0.825 V, and −0.839 V, respectively. On the basis of the open-circuit potential, the scanning range of the potentiometric polarization curve was determined as −1.435 V to −0.235 V. Figure 12 shows the potentiometric polarization curve. The corrosion potential (E_corr_) of the WNZ (−0.765 V) was higher than that of the TMAZ (−1.150 V), which was higher than that of the HAZ (−1.205 V). Higher E_corr_ values indicate better corrosion resistance; thus, the WNZ exhibited the best corrosion resistance of all samples.

According to the corrosion fatigue analysis, the microstructure of the 7075 aluminum alloy FSW joints was different in each region, which resulted in corrosion potential differences. Corrosion potential differences can promote the occurrence of galvanic corrosion. When the whole joint was immersed in 3.5 wt.% NaCl solution, galvanic corrosion occurred at the junction region of the different zones of the joint because of the different corrosion potentials. A greater potential difference results in a greater driving force for galvanic corrosion. The potential difference between the WNZ and TMAZ was 0.385 V, and the difference between the TMAZ and HAZ was 0.055 V. When the potential difference is greater than 0.25 V, galvanic corrosion takes place [28]. Therefore, galvanic corrosion occurred in the junction region between the WNZ and TMAZ, accelerating the local corrosion in this area. By analyzing the polarization curves of each zone in the joint and observing the microstructure, it was then determined that the main crack source of corrosion fatigue was located at the boundary between the WNZ and TMAZ.

### 3.4. Corrosion Fatigue Fracture Morphology

We recorded the interface of the corrosion fatigue fracture information related to the fracture process, from the initiation, propagation, and final fracture of the corrosion fatigue crack of the sample to the material deformation. The main purpose of fracture feature analysis is to determine the source of the cracks [29]. In order to explore the corrosion fatigue fracture process of the aluminum alloy FSW joint, explain the corrosion fatigue fracture mechanism, and determine the reason for failure, the fracture characteristics of the samples were analyzed by scanning electron microscopy.

Figure 13 shows the macro morphology characteristics of corrosion fatigue fracture. Overall, the corrosion fatigue fracture was composed of a crack source region, a crack propagation region, and a transient fracture region. In the sample studied in this report, many crack sources were observed, including a small number of main crack sources and a large number of secondary crack sources. The fracture morphology (Figure 13) shows that the corrosion is more serious at the main crack sources and lighter at the secondary crack source. According to the electrochemical measurements, the main crack source was located at the boundary between the WNZ and TMAZ. The main crack source is typically the first place to produce corrosion pits. In addition, the TMAZ on the upper surface and side of the specimen exhibited multiple corrosion pits, which subsequently developed into small cracks, becoming the secondary crack source. At that time, multiple cracks expanded to the internal WNZ and intersected to form a large crack. When the crack was further extended, the internal force between the grains was not enough to bear the external load, and the crack rapidly expanded to fracture.

Through further observation, it was found that the corners of the sample became arc-shaped after corrosion fatigue (Figure 13) because of the higher stress level and suffered more corrosion damage compared with the plane.

#### 3.4.1. Corrosion Fatigue Crack Initiation

The crack source is the starting point of corrosion fatigue failure. The corrosion fatigue cracks were mainly generated at the corrosion pits on the sample surface and originated from multiple pits. River-like morphologies could be observed around the pits, as shown in Figure 14. On the one hand, corrosion pits lead to stress concentration. On the other hand, corrosion and stress lead to the reduction of the elastic modulus [17], making it prone to crack. The second-phase particles are often the origin of aluminum alloy corrosion [26,30]. In the FSW process, the 7075 aluminum alloy welding joint was subjected to strong agitation and thermal cycling by the agitator needle. A large number of second-phase particles were observed in the microstructure, which might occur near the Al–Cu–Fe–Zn particles, used as the cathode relative to the matrix. Alternatively, Al–Mg–Zn particles were used as the anode relative to the matrix and preferentially corroded to form pitting corrosion in the NaCl solution [31]. In addition, due to the differences in hardness and Young’s modulus of the second-phase reinforced particles and the matrix, the bonding strength of the strengthened particles and the matrix was relatively weak, and the stress and strain concentration occurred around the strengthened particles [17]. Consequently, the second-phase particles or the interface between the second-phase particles and the matrix were also the origin of fatigue cracks [32]. Therefore, the initiation of corrosion fatigue cracks in the process of corrosion fatigue under the coupling action of alternating stress and corrosion was promoted in the particle distribution area of the second phase. In addition, the stress concentration at the corner was enhanced, resulting in the corner being the main crack source.

#### 3.4.2. Corrosion Fatigue Crack Propagation

A previous study [33] showed that the fracture mechanism was a mixed fracture mechanism of ductile and quasi-cleavage fracture in the growth region of the inert environment. However, it can be seen from Figure 15 that tearing ridges appeared at the fracture in the growth region of the corrosion fatigue crack, with intergranular fracture [17] and fatigue speckle characteristics observed between the tearing ridges (Figure 15a). Due to the sensitivity of the 7075 aluminum alloy FSW joint region to intergranular corrosion in 3.5 wt.% NaCl solution, such corrosion occurred at the fracture. Since the precipitation-free zone at the grain boundary is the weakest place in the microstructure [34], fracture occurred in the weak grain boundary area under the action of alternating stress, where cracks easily propagated, thus highlighting the brittle fracture characteristics of the material. In addition, in the process of intergranular corrosion, a “wedge effect” occurred at the grain boundary due to the volume expansion of corrosion products generating tensile stress, which, coupled with alternating tensile stress, led to the expansion of the intergranular crack. Moreover, corrosion channels easily formed under the joint action of corrosion solution and fatigue load due to the low interface energy at the grain boundary, and anodic dissolution occurred, forming the morphological characteristics similar to a “rock candy lump” (Figure 15b). In the process of gradual crack propagation, with the continuous enhancement of corrosion, the expansion area became the most seriously corroded area. Simultaneously, the bearing area of the sample section gradually decreased and the effect of alternating stress on the section gradually increased, resulting in serious damage to the grains and intergranular region, showing the morphological characteristics of an “ant nest” (in Figure 15c).

#### 3.4.3. Corrosion Fatigue Transient Crack

Aluminum alloy has ductility characteristics; however, in the instant fracture zone, it exhibited a large number of cleavage steps (marked with white arrows in Figure 16), showing the characteristics of brittle fracture. This is also different from the mixed brittle and ductile fracture feature when undergoing fatigue in an inert environment [22]. This indicates that the brittle feature of the instant fracture zone was caused by hydrogen embrittlement occurred in corrosion environment. With gradual crack propagation, the alternating load in this area decreased, and the stress exceeded the limit load of the section, resulting in secondary cracks (marked with a red arrow in Figure 16) and transgranular fracture characteristics. In addition, holes appeared in the local area of the transient fault zone, which were caused by the second-phase particles in the WNZ of the FSW joint falling off from the matrix during corrosion fatigue [35].

## 4. Conclusions

In this paper, the corrosion fatigue characteristics of 7075-T6 aluminum alloy FSW joints in 3.5 wt.% NaCl solution were studied. By analyzing the microstructure of the FSW joints, the dynamic potential polarization curve, and the fracture morphology of the corrosion fatigue, the following conclusions were drawn:(1)The S–N curve equation of 7075 aluminum alloy FSW joint was lgN = 5.852 − 0.014S.(2)The difference in the microstructure of the welded joint resulted in different corrosion potentials. After friction stir welding, fine equiaxed crystals were formed in the WNZ, resulting in a high potential. The formation of coarser grains in the TMAZ and HAZ resulted in a lower potential. At the junction of the TMAZ and WNZ, galvanostatic corrosion was generated due to the potential difference, which made this area the most sensitive to corrosion. At this location, corrosion pits were first generated and became the main crack source of corrosion fatigue.(3)Corrosion fatigue fracture is composed of the crack source region, the crack growth region, and the transient fault region. The crack originated at the boundary between the TMAZ and WNZ, which propagated through the WNZ and finally broke in the WNZ.(4)Many crack sources were observed in the corrosion fatigue crack, and the crack sources occurred at the corrosion pit. The crack growth area exhibited fatigue speckles and an intergranular corrosion morphology. Cleavage steps and secondary cracks were observed in the transient fault zone, showing the characteristics of brittle fracture.

## Figures and Tables

**Figure 1 materials-13-04196-f001:**
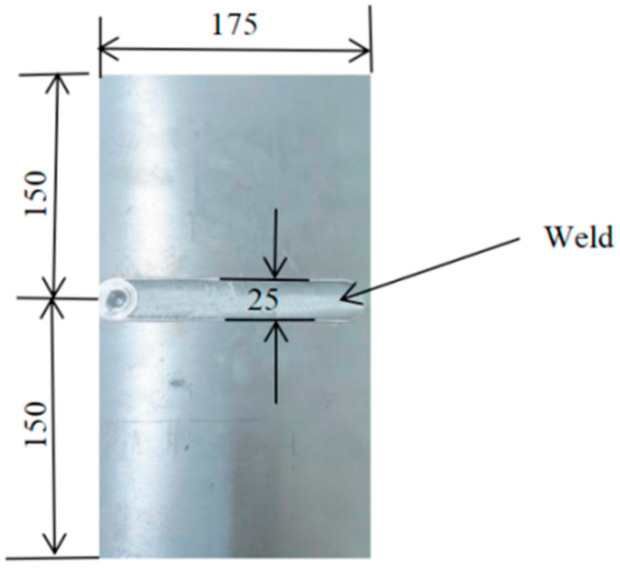
FSW plate (unit: mm).

**Figure 2 materials-13-04196-f002:**
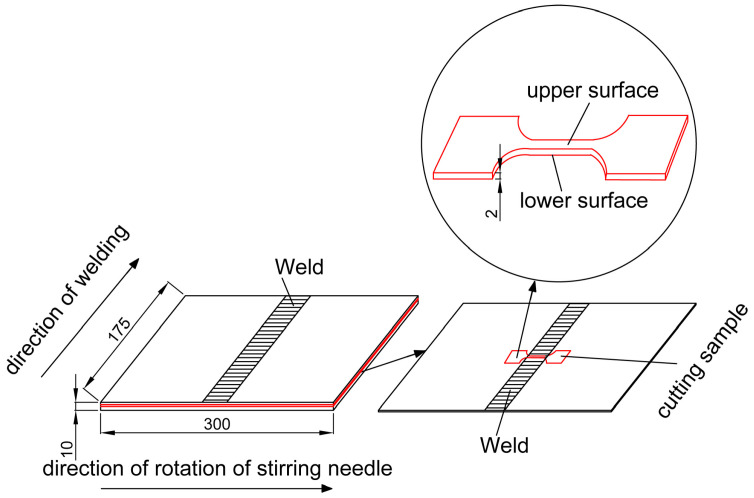
Schematic of the location of the samples.

**Figure 3 materials-13-04196-f003:**
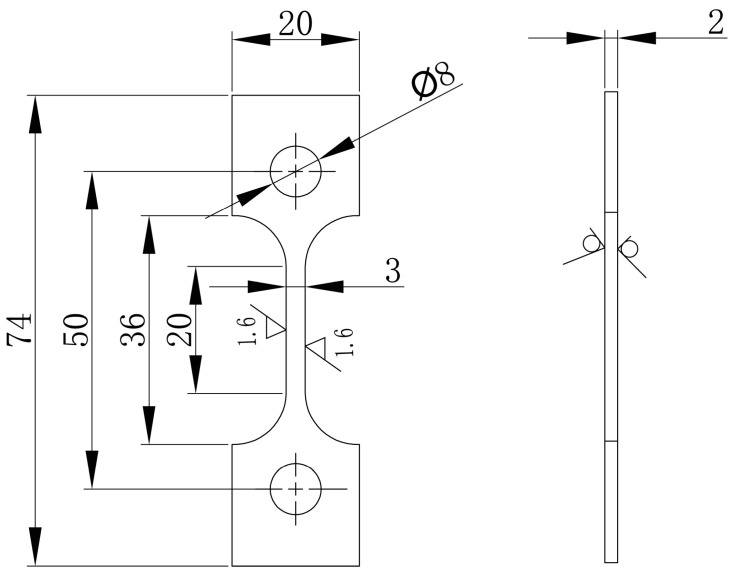
Details of the corrosion fatigue specimen (unit: mm).

**Figure 4 materials-13-04196-f004:**
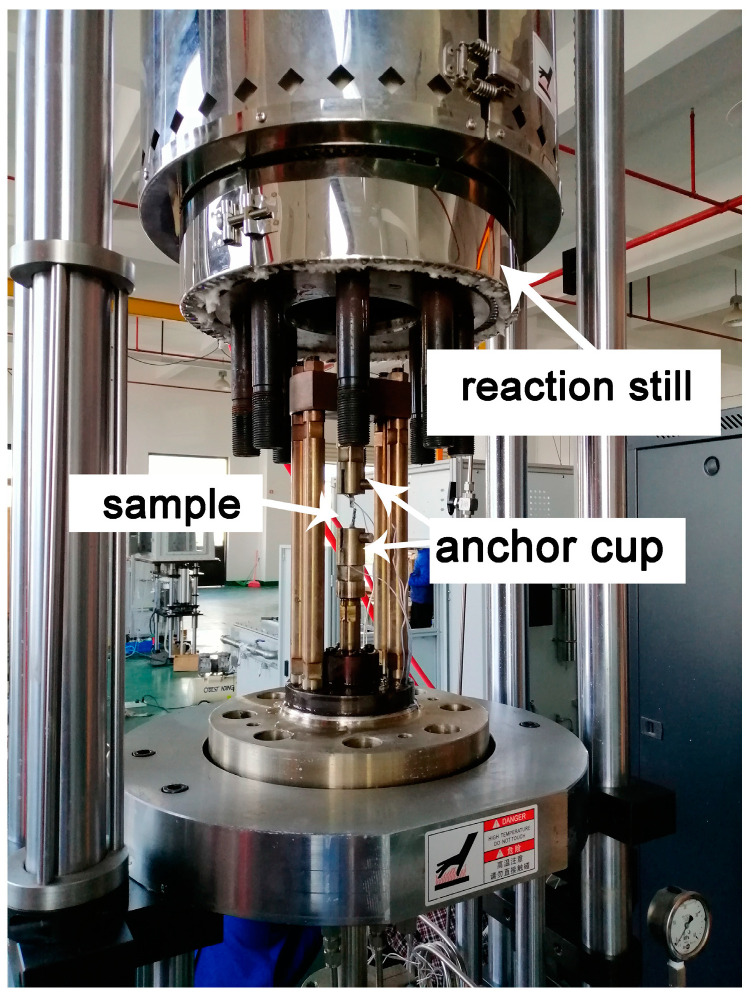
Corrosion fatigue testing machine.

**Figure 5 materials-13-04196-f005:**
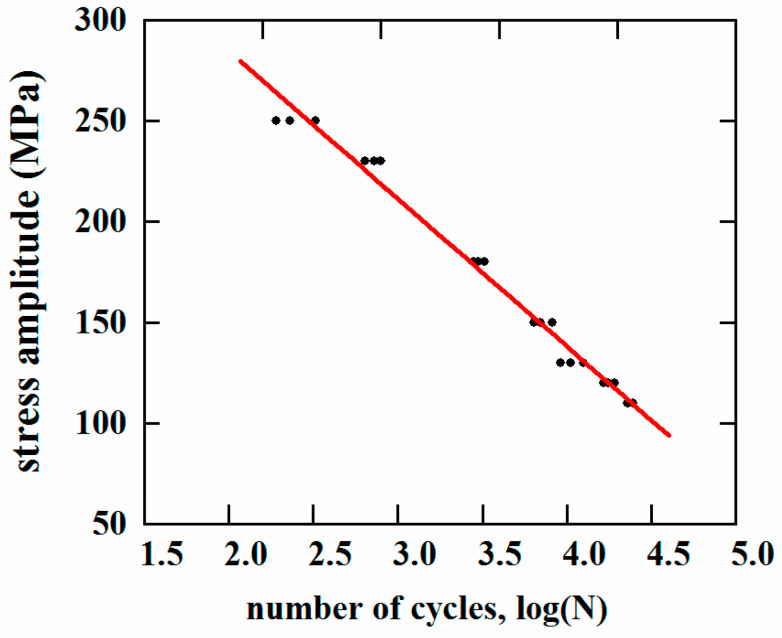
Corrosion fatigue S–N curve of the FSW specimens.

**Figure 6 materials-13-04196-f006:**
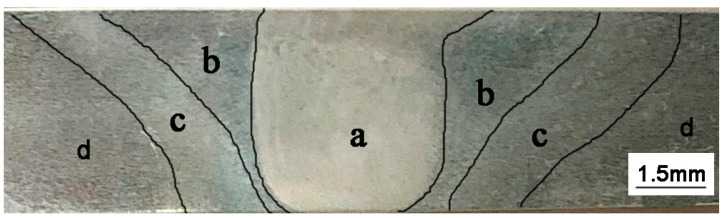
Macrostructure along the cross-section of the FSW: (**a**) weld nugget zone (WNZ); (**b**) thermomechanically affected zone (TMAZ); (**c**) heat-affected zone (HAZ); (**d**) base metal zone.

**Figure 7 materials-13-04196-f007:**
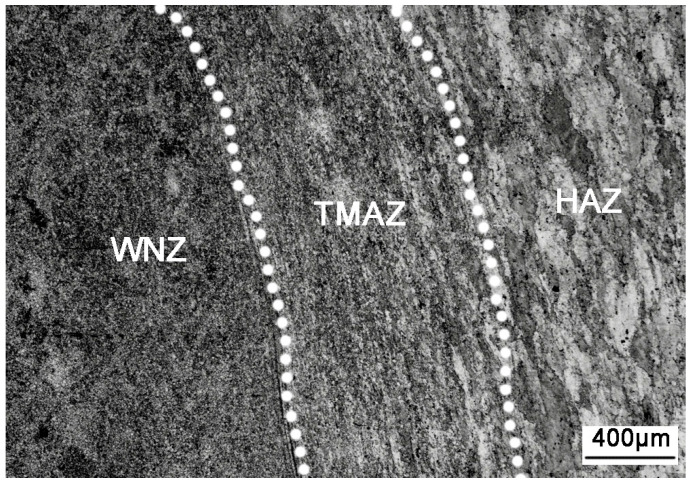
Microstructure of the FSW joint.

**Figure 8 materials-13-04196-f008:**
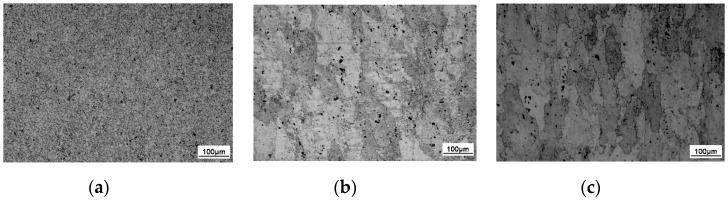
Metallographs of different zones in the aluminum alloy FSW joints: (**a**) WNZ; (**b**) TMAZ; (**c**) HAZ.

**Figure 9 materials-13-04196-f009:**
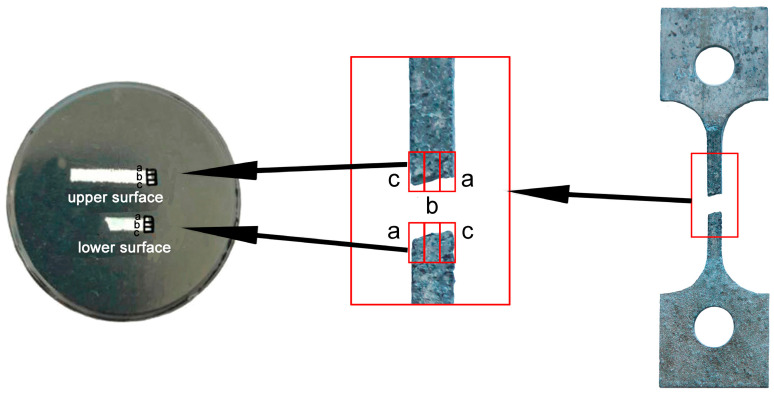
Schematic diagram of the position selected for microstructure observation.

**Figure 10 materials-13-04196-f010:**
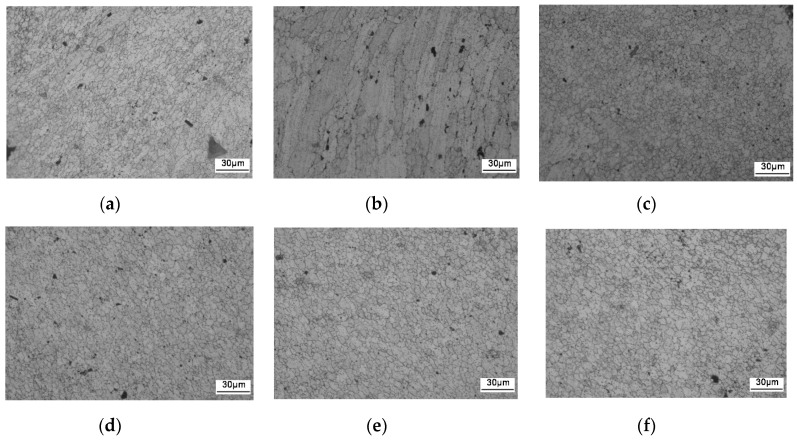
Microstructure of the upper and lower surfaces of the fracture: (**a**) upper surface of area a; (**b**) upper surface of area b; (**c**) upper surface of area c; (**d**) lower surface of area a; (**e**) lower surface of area b; (**f**) lower surface of area c.

**Figure 11 materials-13-04196-f011:**
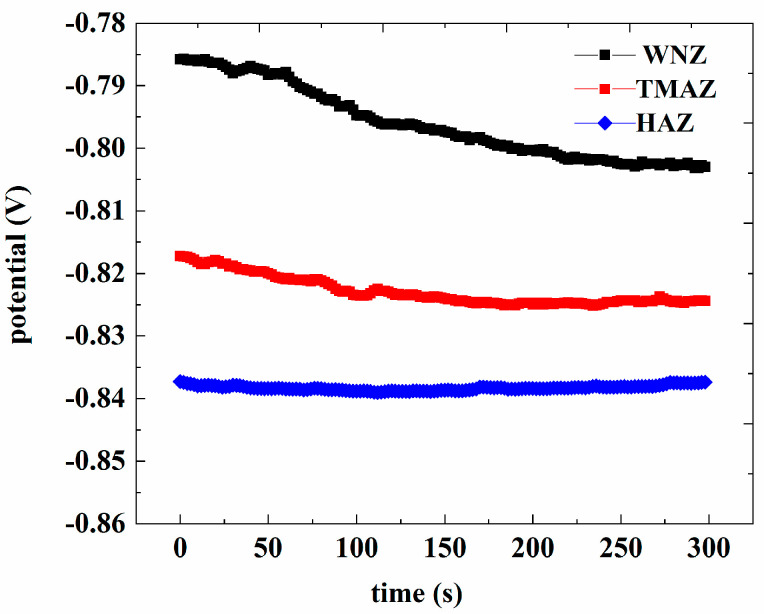
Open potentials of the welded joint in 3.5 wt.% NaCl solution in different positions.

**Figure 12 materials-13-04196-f012:**
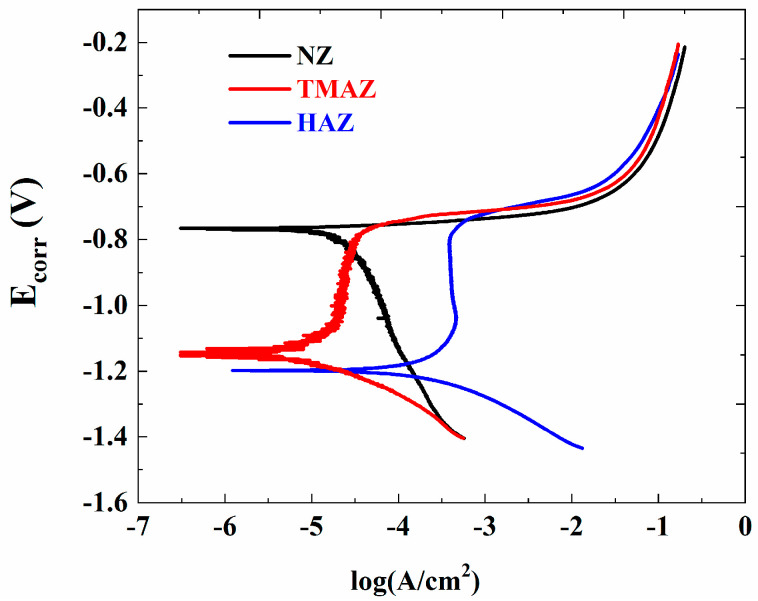
Polarization curves of the joints in 3.5 wt.% NaCl solution in different regions.

**Figure 13 materials-13-04196-f013:**
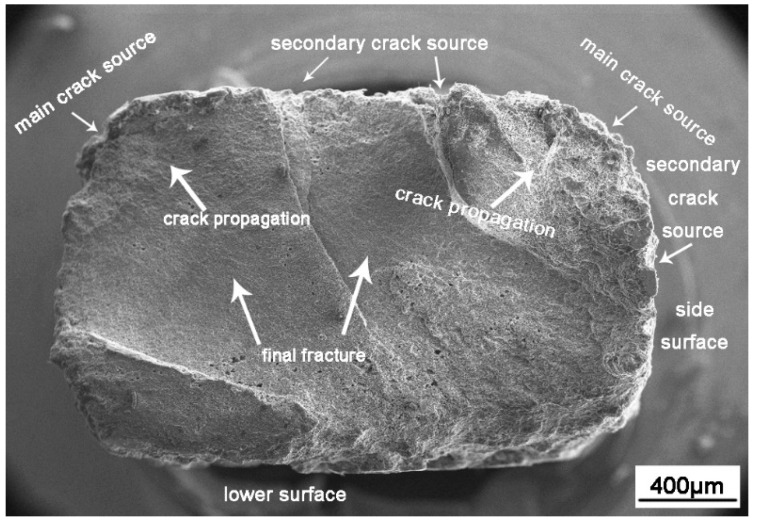
Corrosion fatigue fracture micrograph.

**Figure 14 materials-13-04196-f014:**
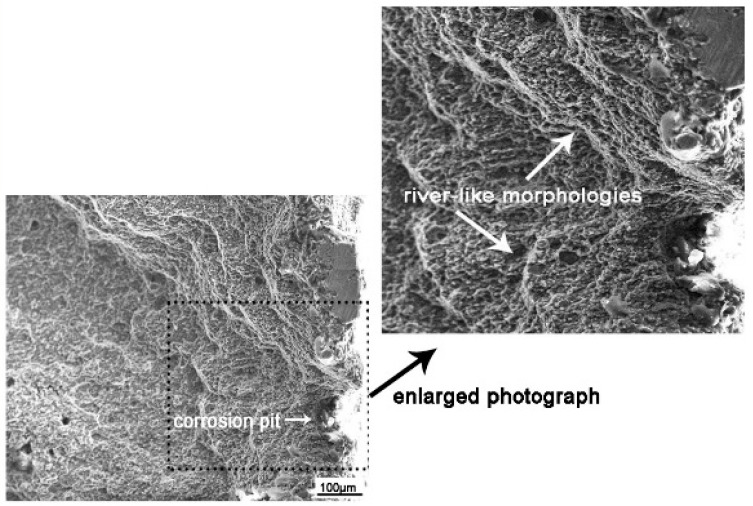
Fracture morphology of corrosion fatigue source.

**Figure 15 materials-13-04196-f015:**
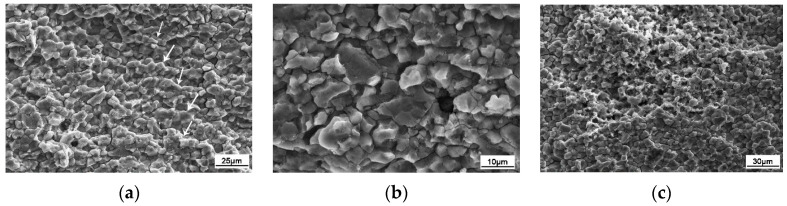
Morphology of the crack propagation zone: (**a**) fatigue striations; (**b**) intergranular corrosion; (**c**) ant-nest corrosion.

**Figure 16 materials-13-04196-f016:**
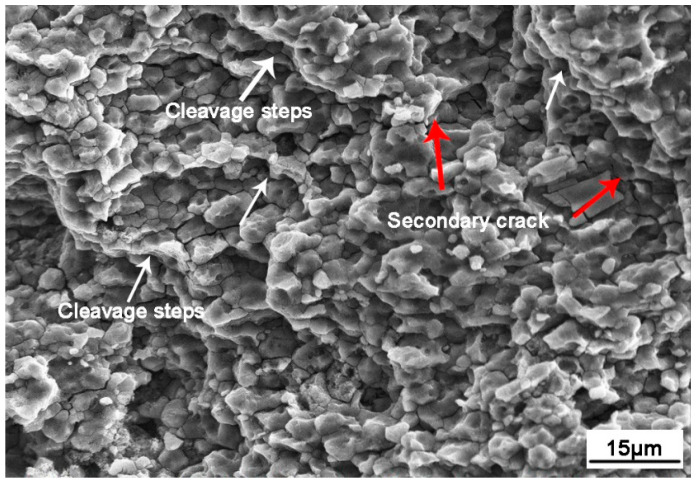
Morphology of the final fracture zone.

**Table 1 materials-13-04196-t001:** Composition of aluminum alloy 7075-T6 (wt.%).

Zn	Mg	Cu	Mn	Ti	Cr	Fe	Si	Al
5.1–6.1	2.1–2.9	1.2–2.0	0.3	0.2	0.18–0.28	0.5	0.4	Balance

**Table 2 materials-13-04196-t002:** Mechanical properties of 7075-T6.

Material	Tensile Strength R_m_ (MPa)	Yield Strength R_p0.2_ (MPa)	Elongation (%)
7075-T6	530	460	6

**Table 3 materials-13-04196-t003:** Welding parameters of FSW

Welding Technology	Welding Machine Model	Rotational Speed n (rpm)	Travelling Speed *ν* (mm·min^−1^)	Tilt Angle of the Rotating Tool *θ* (°)	Rotational Direction of the Rotating Tool
FSW	SN-TS1106-6T-2D	700	150	2.5	Anticlockwise

**Table 4 materials-13-04196-t004:** Corrosion fatigue results of 7075 aluminum alloy FSW joints.

Specimen	Stress Amplitude (MPa)	Loading Frequency (Hz)	Stress Ratio	Fatigue Life (Cycles)	Fracture Location
A1	250	0.3	0.06	190	Weld joint
A2	250	0.3	0.06	230	Weld joint
A3	250	0.3	0.06	326	Weld joint
B1	230	0.3	0.06	645	Weld joint
B2	230	0.3	0.06	732	Weld joint
B3	230	0.3	0.06	790	Weld joint
C1	180	0.3	0.06	2822	Weld joint
C2	180	0.3	0.06	2980	Weld joint
C3	180	0.3	0.06	3243	Weld joint
D1	150	0.3	0.06	6435	Weld joint
D2	150	0.3	0.06	7023	Weld joint
D3	150	0.3	0.06	8230	Weld joint
E1	130	0.3	0.06	9230	Weld joint
E2	130	0.3	0.06	10560	Weld joint
E3	130	0.3	0.06	12542	Weld joint

**Table 5 materials-13-04196-t005:** S–N curve equations and parameters.

Material	S–N Equation	m (−0.014 = −m *lg*e)	C (lg C = 5.852)
7075-T6	lgN = 5.852 − 0.014S	0.032	0.71 × 10^6^

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
