# Peer review of "Corrosion Fatigue Fracture Characteristics of FSW 7075 Aluminum Alloy Joints"

_materials, 2020, doi:10.3390/ma13184196_

Round 1
Reviewer 1 Report
The paper describes an interesting problem regarding the fatigue-corrosion cracking of FSW joints made of aluminium alloy.
Even though the paper contains scientific novelty, its readability (typos, text arrangement) should be improved and then, the article should be re-reviewed. Simply, in current form, the paper is difficult to go through. The paper could be shortened and rewritten to be more concise. My comments on the article:
- In material and methods, the geometry of the tool should be presented.
- Authors write " The samples were polished using silicon carbide paper up to 1200 grit 93 before corrosion fatigue test. " - the geometry of the samples should be explained. Which surfaces were polished? The whole four plates of dog bone sample? Please explain.
- In the fig. 2 the roughness in the right sketch should be given.
- I am worried about the geometry of the samples used for fatigue tests. Samples were machined to obtain a 2 mm thickness. Therefore almost 8 mm of the sample was removed. It is not clear if the sample was taken from the middle of the 10 mm plate? Especially while the weld was wide for 25 mm (see fig 1). Please provide details of the dog bone location in the as-welded sample.
- The samples numbering is not clear. Please add the table to explain the sample codes. (section 2.2).
- This phrase is not clear "Because the fracture
108 position of the corrosion fatigue fracture will change after polishing. " - please explain it. - There are too many phrases "Error! Reference source not found.." therefore it is really difficult to go through the manuscript. It must be improved.
- All the literature references are placed in the text as a "free numbers" it must be improved and placed in square brackets.
- Macrostructure given in fig 4 should present whole joint cross-section. Therefore, more photos are needed.
- Figure 7 is not clear. The reader does not understand the meaning of the arrows, description and "round element on the left". Please improve it.
- figure 8 - I think that the microstructure is much more affected by the location of the sample, and the area of the FSW joint from which the sample was machined.
- Explain in the paper the meaning of the phrase "the upper
183 and lower surface " at its first usage in the text. - Authors write that " river-like morphologies around the
272 pits " - but it is not visible in the SEM photos. Please provide the SEM with higher magnification and mark it in the fracture-photo. - The difference between the secondary and primary cracks is not clear and I am not convinced with that - please explain it. Please provide the explanation for distinguishing the types of cracks.
- The conclusions are difficult to evaluate because the paper content is blurred.
Reviewer 2 Report
Review
The manuscript presents a study on corrosion fatigue properties and fracture characteristics of friction stir welded joint of the 7075 aluminum alloy. In general, the idea for the study is interesting, both from the scientific and application point of view. Nevertheless, some issues have to be addressed before the manuscript is considered for publication:
- The English language is good enough for the review process but it is far away from being acceptable for a decent article to be published in MDPI materials, therefore it needs extensive corrections both technically and stylistically.
- The style of references should be the same throughout the text.
Other important issues that should be addressed are given by line:
Lines 90–91: Authors should illustrate the specimen extraction position visually in Figure 1 or 2, or at least clarify textually because the sentence “…cut by linear cutting along the vertical direction of the weld center of the butt welding plate …” is not clear enough!
Lines 173–206: Images in Figures 4, 5, 6, and 8 are optical micrographs, but optical microscopy was not mentioned in the “Materials and methods” section. Please make sure you add a sentence about the optical microscopy. In addition, two scanning electron microscopes (SEM) were mentioned. Which ones are the secondary electron images from?
Line 140 onward: The whole “Results and discussion” section has an error in references to figures and tables. Although this is just a technical issue, it should have been checked before the manuscript was sent for review, because this makes the review process much more difficult!
Line 238 and 240: Figures 8 and 9 should have the same style of unit representation in abscissa and ordinate. Please check and correct accordingly.
Line 263: The white text in Figure 10 should be of the same size and not deformed.
Reviewer 3 Report
The present contribution seems to be not cross-checked by the authors and needs some revision before publication in Materials.
- Whole paper: Check the whole paper carefully if there are redundant spaces which must be deleted. Insert spaces where necessary (especially between numerical value and unit, etc.).
- The arrangement between text and figures/tables has to be optimized (see page 2, 4 and 10).
- Line 16/17: “Corrosion fatigue cracks exist multiple source regions.” – Not clear.
- No capitalization: “Welding” (line 32 and 34), “And” (line 94), Wt.%” (line 296) and “As” (line 310).
- Table 2: “Rm” and “Rp0.2” – Define where first used. “m” and “p0.2”: subscript.
- Table 3: Capitalization: “welding” and “rational”. “-1”: superscript. Symbols in the same line than the rest of the text.
- Line 121: “cm2” – Superscript “2”.
- “Error! Reference source not found.”: Figure are nor referred (line 140, 145, 147, 149, 156/157, 157, 164/165, 169/170, 171/172, 184, 186/187, 216, 219, 252, 266, 272, 292, 294/295, 305 and 318).
- Line 148: Test constants M and C are not well-defined. “M” was not used but “m” (see Table 5”.
- Table 5: Similar font size.
- Figure 3: Too small, font size much too small.
- Numbers in the text – references?: “22-24” (line 156), “25” (line 163), “26” (line 164), “27” (line 171), “28” (line 233), “29” (line 248), “17” (line 247), “26, 30” (line 275), “31” (line 280), “17” (liner 283), “22” (line 284), “17” (line 294), “33” (line 296) and “34” (line 323).
- Line 208: Replace “Figure 8” with “Figure 7” (see page 8 for comparison).
- Line 220/221: Define “Ecorr” where first used. Subscript “corr”.
- Numbers in the text – chapters?: “3.1” (line 224) and “3.3” (line 256)
- Figure 9: Too small, font size much too small. Figure caption on the same page.
- Line 278: Replace “al-Mg-Zn” with “Al-Mg-Zn”.
- References must be according to the instructions to the authors.
Round 2
Reviewer 1 Report
Generally, I accept the explanations given by the authors. The paper was improved however I still have some comments on the paper content.
- Please remove the figs. 5,6,7 - it is unnecessary to publish in such a prominent journal as Materials the photos of basic-materials science equipment.
- please show in fig 4 how the sample was immersed in a 3.5 wt. % NaCl solution. Add the photo of the sketch. How it was sealed?
- In fracture analysis: please comment that the identified fracture is different from those reported for "classic cleavage and river pattern" observed for the steel samples, see the literature https://doi.org/10.1016/j.engfailanal.2020.104447 I think it should be stated or commented on in the text, that the non-ferrous materials present ductility and usually exhibit brittle fracture, see the literature https://doi.org/10.1007/s11661-013-2013-3. Maybe in your case, it relies on the microstructure and can be connected with the coarse-grain weld structure - please consider it.
